# The Association of Chronic Pulmonary Aspergillosis and Chronic Pulmonary Histoplasmosis with MDR-TB Patients in Indonesia

**DOI:** 10.3390/jof10080529

**Published:** 2024-07-29

**Authors:** Noni N. Soeroso, Lambok Siahaan, Selfi Khairunnisa, Raden Ajeng Henny Anggriani, Aida Aida, Putri C. Eyanoer, Elvita R. Daulay, Erlina Burhan, Anna Rozaliyani, Ronny Ronny, Robiatul Adawiyah, David W. Denning, Retno Wahyuningsih

**Affiliations:** 1Department of Pulmonology and Respiratory Medicine, Faculty of Medicine, Universitas Sumatera Utara, Universitas Sumatera Utara Hospital, Medan 20155, Indonesia; noni@usu.ac.id (N.N.S.); skhairunnisa11@gmail.com (S.K.); anggrianihenny@gmail.com (R.A.H.A.); ayamsemur@gmail.com (A.A.); 2Department of Parasitology, Faculty of Medicine, Universitas Sumatera Utara, Medan 20155, Indonesia; lamboksiahaan_fkusu@yahoo.com; 3Department of Community and Preventive Medicine, Faculty of Medicine, Universitas Sumatera Utara, Medan 20155, Indonesia; putrieyanoer@usu.ac.id; 4Department of Radiology, Faculty of Medicine, Universitas Sumatera Utara, Medan 20155, Indonesia; elvitarahmi.daulay@usu.ac.id; 5Department of Pulmonology and Respiratory Medicine, Faculty of Medicine, Universitas Indonesia, Jakarta 13230, Indonesia; erlina_burhan@yahoo.com; 6Department of Parasitology, Faculty of Medicine, Universitas Indonesia, Jakarta 10430, Indonesia; anna.rozaliyani@ui.ac.id (A.R.); robiatul.adawiyah01@ui.ac.id (R.A.); 7Lung Mycosis Centre, Department of Parasitology, Faculty of Medicine, Persahabatan Hospital, Universitas Indonesia, Jakarta 13230, Indonesia; 8Department of Parasitology, Faculty of Medicine, Universitas Kristen Indonesi, Jakarta 13630, Indonesia; ronny@uki.ac.id; 9Clinical Parasitology Study Programme, Faculty of Medicine, Universitas Indonesia, Jakarta 10430, Indonesia; 10Manchester Fungal Infection Group, The University of Manchester, Manchester Academic Health Science Centre, Manchester M13 9PL, UK; ddenning@manchester.ac.uk

**Keywords:** multi-drug resistant tuberculosis, anti-*Histoplasma* IgG antibody, anti-*Aspergillus* IgG antibody, infectious disease

## Abstract

In Indonesia, 2.4% of all new tuberculosis patients had multi-drug resistant disease (MDR-TB); an estimated 24,000 incidences. Historical case series of MDR-TB described a high frequency of cavitation and poor prognosis. The diagnosis of chronic pulmonary aspergillosis (CPA) relies on raised levels of *Aspergillus* IgG antibodies, and detectable *Histoplasma* IgG antibodies are suspicious for chronic pulmonary histoplasmosis (CPH). We investigated whether MDR-TB patients might have concurrent CPH or CPA. This was a cross-sectional study with 50 MDR-TB patients. ELISA was used to detect *Histoplasma* IgG antibodies and lateral flow assay was used to detect *Aspergillus* IgG/IgM antibodies. Several other possible disease determinants were assessed by multivariate analysis. Of the 50 MDR-TB patients, 14 (28%) and 16 (32%) had positive *Histoplasma* or *Aspergillus* serology; six patients (12%) had dual antibody reactivity. Radiological abnormalities in positive patients included diffuse or local infiltrates, nodules, consolidation, and apical cavities, consistent with CPH and CPA. Patients with detectable fungal antibodies tended to have worse disease, and 4 of 26 (15.3%) died in the first 5 months of dual infection (*p* = 0.11 compared with no deaths in those with only MDR-TB). The criteria for the diagnosis of CPH and CPA were fulfilled in those with moderately and far advanced disease (13 of 14 or 93%) and 12 of 16 (75%), respectively. Damp housing was the only determinant associated with *Histoplasma* antibodies (PR 2.01; 95%CI 0.56–7.19), while pets were associated with the *Aspergillus* antibody (PR 18.024; 95%CI 1.594–203.744). CPA or CPH are probably frequent in MDR-TB patients in Indonesia and may carry a worse prognosis.

## 1. Introduction

The Global Tuberculosis Report from the World Health Organization estimated that 10.6 million people suffered from tuberculosis (TB) in 2021, an increase of 4.5% from 2020. Indonesia has the third highest incidence of tuberculosis in the world after India and China, according to the Global Tuberculosis Report in 2021 [1,2].

Basic health research data showed the incidence of TB in Indonesia to be 316 per 100,000 population in 2018, with an estimated 845,000 active cases. The case notification rate of all TB cases in Indonesia varies between 69 and 268 per 100,000 population by province, in 2021, with Papua province having the highest caseload and Bali the lowest. North Sumatra province had 150 cases per 100,000 population in 2021 [3].

Based on the TB case data reported in 2021, multidrug-resistant tuberculosis (MDR-TB) or rifampicin-resistant tuberculosis (RR-TB) represented 3.6% of new cases and 18% of previously treated cases. Three countries accounted for 42% of global MDR cases in 2021: India (26%), Russian Federation (8.5%), and Pakistan (7.9%). In Indonesian patients, 2.4% of all new tuberculosis cases had MDR-TB with 13% of all ongoing treatment or treated TB cases, with an estimated 24,000 incidence [2].

Multidrug-resistant tuberculosis is defined as resistance to rifampicin and isoniazid (RH), with or without other first-line drugs [4]. Rifampicin-resistant tuberculosis can be defined as TB caused by strains of *Mycobacterium tuberculosis* that are resistant to rifampin, but these strains may be resistant to isoniazid or resistant to other first-line or second-line TB drugs. Therefore, MDR-TB and RR-TB cases are often lumped together as MDR/RR-TB and are eligible for treatment with MDR-TB regimens [5].

The clinical symptoms of MDR pulmonary TB include a history of chronic cough and non-specific symptoms such as anorexia, lethargy, and fever. As the disease progresses, the cough becomes productive, with purulent and sometimes hemoptysis, pleuritic chest pain, shortness of breath, night sweats, and weight loss [6]. The general symptoms of pulmonary mycoses such as chronic pulmonary histoplasmosis (CPH) and chronic pulmonary aspergillosis (CPA) are similar to other microbial infections including coughing or coughing up blood, a lot of phlegm, shortness of breath, fever, and chest pain, but they can also be asymptomatic [7,8,9]. It is difficult to distinguish from TB or MDR-TB from other chronic infections such as CPA and CPH. Both conditions can damage the lung parenchyma and cause progressive cavitation, pleural thickening, and fibrosis. Radiologic features such as cavitation, peri-cavitary fibrosis, and pleural thickening were present in 67–80% of cases of CPA in a recent study from Ghana [10]. These manifestations are also relatively similar to CPH.

In a recent study of CPA in Indonesia of 216 patients diagnosed with pulmonary TB, 12 (6%) patients met the criteria of proven CPA, 5 (2%) patients as probable CPA, and 15 (7%) patients as possible CPA. *Aspergillus*-specific IgG was positive in 64 (30%) patients and an additional 16 (13%) patients sero-converted from negative to positive during TB therapy [11].

The mapping of histoplasmosis in Southeast Asia by Baker et al., [12] reported a total of 407 cases from Thailand (233 cases), Malaysia (75 cases), Indonesia (48 cases), and Singapore (21 cases). Tanjung et al. [13]. reported 13.6% cases with a positive histoplasmin skin test performed in Medan. In the 1950s, a study in the United States reported 7.2% of patients in the Missouri State TB sanatorium were cases of CPH [14]. Meanwhile, data regarding CPH in Indonesia can be said to be non-existent.

Data on CPA among MDR-TB cases are still limited; one study in Brazil found three of 48 (6%) with CPA and four with chronic pulmonary paracoccidioidomycosis [15]. No other data are series are published, and none on CPH in MDR-TB patients. The aim of this study was to compare the prevalence of CPH and CPA, their clinical characteristics, radiological features, clinical outcomes and other supporting data in MDR-TB patients in Medan, North Sumatra. This is the first study to examine the incidence of CPA as a co-infection of MDR-TB patients in Indonesia in tandem with anti-*Histoplasma* IgG and CPH.

## 2. Materials and Methods

Study design

This was a study of cross-sectional design, with consecutive sampling conducted at Adam Malik Hospital in the Department of Pulmonology and Respiratory Medicine, Faculty of Medicine, Universitas Sumatera Utara from July 2021 to September 2021. Some of the data were previously reported for anti-*Histoplasma* IgG antibodies [16]. The entire research procedures and steps were carried out in accordance with the Declaration of Helsinki and authorized by the Research Ethics Committee of the Faculty of Medicine, Universitas Sumatera Utara, Indonesia, and Faculty of Medicine Universitas Indonesia. All the subjects (patients and guardians) signed a written consent form. The research experimental protocol was carried out in strict accordance with the applicable guidelines.

Subjects

Patients who participated in this study were diagnosed with MDR-TB and were treated regularly at the outpatient infectious diseases clinic at Adam Malik hospital, Medan, North Sumatra. Inclusion criteria were patients aged >18 years old with a confirmed diagnosis of multidrug-resistant TB based on rapid molecular and MGIT tests and having complete medical records. Patients demonstrated willingness to participate in the study by signing the informed consent forms. Exclusion criteria were patients with lung TB diagnosis with rapid molecular tests that showed rifampicin sensitivity, incomplete medical records, and those unwilling to participate in the study. Data collected included demographic, clinical characteristics, laboratory data, TB therapy data, comorbid diseases, smoking habits, work history in agriculture, place of residence, and radiological data of the lungs.

Chest X-ray abnormalities were classified [17] as (1) minimally advanced—slight to moderate lesion density with no cavitation, unilateral and limited to between the second chondro-sternal junction and spine of the fourth thoracic vertebrate; (2) moderately advanced—extensive soft infiltrates in one lung or bilateral or dense confluent lesions limited to one-third of the volume of one lung and, if cavitation present, less than 4 cm in diameter; (3) far advanced—more extensive lesions than moderately advanced image.

Rapid Molecular Test for Mycobacterium Tuberculosis

Xpert MTB/RIF and Xpert Ultra are used in Indonesia as the initial diagnostic test for TB and rifampicin resistance (Cepheid, Sunnyvale, CA 94089 USA) [18]. The GeneXpert instrument uses an automated system that integrates specimen purification, nucleic acid amplification, and detection of the target sequences. The system consists of the GeneXpert instrument, a computer, and software (Xpert^®^MTB/RIF Ultra version 4.0). Every test uses a one-time-use cartridge that minimizes cross-contamination. The result is interpreted as “MTB detected” when two probes provide a Ct value within the valid range and a minimum delta Ct min (the smallest Ct difference between any pair of probes) of <2.0. “Rifampicin resistance not detected” is represented by a delta Ct max (the Ct difference between the earliest and latest probes) of ≤4.0. “Rifampicin resistance detected” is reported when delta Ct max is >4.0. “Rifampicin resistance indeterminate” is reported when the two following conditions are met, such as the Ct value of any probe exceeds the maximum valid Ct value (or value is 0) and the Ct value of the earliest probe is >valid Ct max—delta Ct max cut-off 4.0. “MTB not detected” is reported if there is only one or no positive probe. This gen expert examination agrees strongly with the results of rifampin resistance that can already diagnose MDR-TB. The sensitivity level of the TCM tool reached 98.3% and the specificity level reached 99% [4,19].

M-GIT

Every patient with MDR-TB underwent liquid culture examination for *Mycobacterium tuberculosis*. Sputum from each patient was collected and MGIT test was carried out in the Microbiology Laboratory, Adam Malik Hospital, Medan. Sensitive control isolate and resistance control isolate of second line antituberculosis drugs were conducted in accordance with the WHO standard.

Blood and sera

A total of 5 mL of blood was collected to be separated as serum (3 mL) and the remainder was planted on Sabouraud dextrose agar. The serum obtained was stored at minus 80 °C before being tested for the *Histoplasma* and *Aspergillus* antibodies.

Histoplasma antibody detection

The detection of IgG antibodies in samples was carried out by using *Histoplasma* antibody enzyme immunoassay (EIA) kit (MiraVista Diagnostics, Indianapolis, IN, USA). The cut-off values used were <8.0 EU (negative); intermediate: 8.0–9.9 EU (intermediate) and >10.0 (positive). The limit of quantification was >80.0 EU.

Aspergillus ICT IgG-IgM

*Aspergillus* antibodies were detected using commercial immune chromatography technology (lateral flow assay), allowing the simultaneous detection of both IgG and IgM class anti-*Aspergillus* antibodies in human sera (LDBio, Lyon, France). A serum sample of 15 μL was dripped onto the rapid test stick, to which was added 4 drops of the reagent provided. The control and test lines were read at 20 min if positive, and up to 30 min, if negative. All results were determined visually.

Diagnosis of CPA and CPH

CPA was diagnosed according to the GAFFI algorithm, requiring a combination of symptoms persisting for at least 3 months, typical radiological features of cavitation, pericavitary infiltrates and/or pleural thickening and a positive *Aspergillus* IgG antibody test [20]. Consensus criteria for the diagnosis of CPH have not been published, but we adopted the same diagnostic principles that have been applied, with less stringency about the radiological features, as these are less well defined for CPH [21].

Statistical analysis

Data analysis was conducted using inferential statistics to determine the risk factors for histoplasmosis and aspergillosis including multivariate analysis and backward selection logistic regression. Fisher exact was used for some 2 × 2 comparisons. The results with *p* value < 0.05 were significant.

## 3. Results

The number of MDR-TB patients studied was 50. The anti-*Histoplasma* IgG antibody was found in 14 (28%) patients and 36 were negative, including two with indeterminate results [16]. There were 16 (32%) patients with detectable anti-*Aspergillus* IgG/IgM antibodies. A total of six patients had both *Histoplasma* and *Aspergillus* antibodies simultaneously (Table 1). Therefore 20 out of 50 (40%) MDR-TB patients studied may have concurrent fungal disease with either or both fungal pulmonary pathogens.

The demographic characteristics of the patients with respect to antibody status are shown in Table 2. Male sex was more frequently found in the recruited MDR-TB patients, and this is reflected in the frequency of antibody positivity. Diabetes was relatively common among those with detectable antibody—50% of those with *Histoplasma* and 63% of those with *Aspergillus* antibody. None of the patients with detectable antibodies to *Histoplasma* or *Aspergillus* was a current smoker and none had HIV infection.

Patients with detectable antibodies tended to have worse disease. So, 8 of 14 (57.1%) of those with positive *Histoplasma* antibodies had far advanced radiological appearances as did 8 (50%) of those with *Aspergillus* antibody (Figure 1). However, there was no statistical association between severity of chest X-ray appearance and having a positive *Histoplasma* or *Aspergillus* antibody result (*p* = 0.069 and 0.48, respectively). The criteria for the diagnosis of CPH and CPA were fulfilled in those with moderately and far advanced disease (13 of 14 (92.9%) and 12 of 16 (75%), respectively).

There were no deaths in the MDR-TB group who did not have antibodies against either *H. capsulatum* or *Aspergillus fumigatus*. In the group with a positive *H. capsulatum* antibody result, one person died two months after diagnosis of MDR-TB and a month after detection of *Histoplasma* antibodies. Two individuals in the group with positive *Aspergillus* antibodies died two and five months after MDR-TB was diagnosed. Furthermore, one individual who had both antibodies also died three months after the diagnosis of MDR-TB was established (Figure 2). All patients passed away during treatment for MDR-TB; two of them had diabetes mellitus. All four patients had advanced radiological images according to the established criteria, with the female patient who tested positive for both *Aspergillus* and *Histoplasma* antibodies having the most severe lung radiological images compared to the other three patients. That patient had extensive infiltrate in both lungs, and the right lung had collapsed (Figure 1a). Meanwhile, three other patients, although they had extensive infiltrate in both lungs, had no collapsed lungs. The proportion dying with positive fungal antibodies was not significant compared with those without antibodies (*p* = 0.11).

Blood samples for antibody testing were taken one month to 21 months after the diagnosis of MDR-TB was established. Based on the results, patients could be divided into two clusters, i.e., early onset and later onset. Early-onset patients are patients who are positive for antibodies at 1–7 months (Figure 2). No discernible pattern of timing was obvious with regard to anti-*Histoplasma* or anti-*Aspergillus* antibody.

The fungal culture test from all patients did not show fungal growth in any blood samples; respiratory fungal culture was not carried out. An interesting difference between those with positive *Histoplasma* antibodies is the lower frequency of a positive *M. tuberculosis* culture (4 of 14 (28.6%)) compared with *Aspergillus* antibody (10 of 16 (62.5%)) (*p* = 0.06). This finding implies that there are more dual CPA and MDR-TB cases than CPH which may be a more frequent post-MDR-TB sequela, assuming that the GeneXpert signal is low in these patients.

Multivariate analysis by backward selection in Table 3 shows that humid houses, previous TB history, and gender are slightly more common in those with detectable Histoplasma IgG, but this was not statistically significant (PR > 1, 95% C.I. for PR). Humid living spaces increased the risk of histoplasmosis by over two times compared to dry living spaces. In Table 4, a multivariate analysis for positive *Aspergillus* IgG/IgM antibodies shows that a significant variable is pets at home with an eighteen times greater chance of positivity.

Radiological findings showed that men with positive *Histoplasma* antibodies experienced more severe lung damage than women, whereas, in the positive *Aspergillus* antibody group, the numbers were found to be almost similar between men and women (*p* > 0.05). In individuals with antibodies to both *Aspergillus* and *Histoplasma*, lung damage was found to be almost the same in both men and women (Table 5).

## 4. Discussion

The results of this study show that one third of MDR-TB patients have anti-*Aspergillus* or anti-*Histoplasma* IgG antibodies or both. This is most consistent with concurrent or sequential CPA and CPH in MDR-TB patients. The presence of anti-*Aspergillus* IgG antibodies usually signifies CPA in association with compatible (and usually typical) radiological findings [20].

Sputum culture is 20–30% sensitive for CPA (and not specific), whereas IgG antibody is 68–92% sensitive, depending on the assay and context. There are fewer data published analyzing the sensitivity and specificity of anti-*Histoplasma* IgG antibodies, or the performance of other assays such as serum or respiratory sample antigen, respiratory sample PCR or culture or blood culture. In this study, the blood culture for fungi was negative; other tests were not carried out. IgG antibody testing cannot differentiate between active infection and past infection, but the presence of these antibodies at a minimum indicates the body’s response to *Histoplasma* inhalation, as they are quite specific, with the one exception of disease caused by *Blastomyces* spp., which are not known to be present in Indonesia. It would have been desirable to have carried out a prolonged sputum fungal culture to confirm active infection in at least some of these patients. However, the presence of anti-Histoplasma IgG antibodies associated with clinical symptoms and progressive radiological findings is most in keeping with CPH. It is known that CPH has symptoms similar to pulmonary TB; ultimately that clinically CPH and PTB cannot be differentiated.It is known that CPH has symptoms similar to pulmonary TB; ultimately that clinically CPH and PTB cannot be differentiated. A major gap in the literature is the lack of consensus criteria for the diagnosis of CPH [21]. Our findings are most consistent with ~40% incidence of CPA and CPH in MDR-TB patients in Medan, Indonesia. Once confirmed and replicated in other locations, our findings have profound significance for the management of MDR-TB.

We believe this association has only been previously reported once in three patients from Brazil and one from India and never for CPH and MDR-TB [22]. Dual or triple infection with MDR-TB and pulmonary fungi is clearly linked to worse disease radiologically, in this cohort. It may also carry a worse overall prognosis as the deaths during MDR-TB were only seen in those with a positive fungal antibody. As some patients were only available for testing many months after MDR-TB diagnosis, some early deaths could have been missed. Several anti-tuberculous agents have profound interactions with itraconazole and other oral antifungal agents used for treating CPA and CPH, including rifampicin, isoniazid, pyrazinamide and bedaquiline [23].

*Histoplasma* is endemic in Medan Indonesia. A study in Medan conducting histoplasmin tests on 169 medical students found 23 (13.6%) who had positive histoplasmin test results [13]. Dual CPH and CPA has rarely been reported previously. In a series of patients with an aspergilloma seen on chest X-ray in Ohio and from the pre-1960 literature, four of 58 (7%) had CPH as the underlying cause of the cavity [15,19,23,24].

*Aspergillus* spp. is the most common cause of fungal lung infection, including CPA. As with most exposure to *Histoplasma*, healthy individuals rarely become ill [25,26,27]. However, in immunocompromised patients or older people, fungal exposure can present as a serious lung disease, with a wide range of clinical manifestations [28]. HIV-AIDS patients with a CD4 cell count of less than 200 cells/µL develop progressive, severe disseminated histoplasmosis, often misdiagnosed as TB. Equally CPA is often mistaken for pulmonary TB [29].

The *Aspergillus* IgG/IgM assay has been evaluated in several studies including one from Indonesia [30]. Sensitivity and specificity vary from 68 to 92% and 70 to 98%, depending on the country, population studies, availability of CT scanning to assist or rule out the CPA diagnosis and control population [31,32,33,34]. The *Histoplasma* IgG antibody assay we used has been evaluated only in the USA with a small number of CPH patients with an 84% sensitivity and no specificity data [35]. More evaluation has been carried out in cats than humans with CPH, although its performance in acute pulmonary histoplasmosis was better than immunodiffusion and complement fixation at 87.5% sensitivity and a specificity of 95%, using healthy and clinical controls [36,37].

One key question in these cases is whether the CPA or CPH followed the pulmonary TB (possibly because of extensive cavitation), or occurred synchronously, and for CPH, there is a possibility of primary infection. Dual pulmonary TB and CPA is documented, but is probably uncommon, and the diagnosis is difficult because of overlapping symptoms and radiological features [38]. Dual infection with *H. capsulatum* and *M. tuberculosis* are rarely documented since they may present similarly and be misdiagnosed [28]; indeed, clinicians are exhorted to exclude other infections when making a diagnosis of MDR- or XDR-TB [39,40]. There is an extensive body of published work documenting many aspects of CPA, including diagnosis and diagnostic criteria, but much less for CPH [21]. The timing of *Aspergillus* and *Histoplasma* antibody appearance needs further study, but we found an early cluster and a later-onset cluster. We hypothesize that the early-onset cluster was infected with *Histoplasma* or *Aspergillus* from the start, possibly linked to the high frequency of pulmonary cavitation in MDR-TB, at least in the case of *Aspergillus* co-infection. The later-onset group could also be related to extensive cavitation.

The similarity of the clinical characteristics between pulmonary TB and histoplasmosis and the nature of *Histoplasma* as a true pathogenic fungus could allow infection without it being recognized as histoplasmosis. CPH is most common in older patients with preexisting lung diseases, such as emphysema or pulmonary TB, or smoking history [25,26,41]. This is consistent with the findings in this study, in which most MDR-TB with histoplasmosis patients were in the 46–65 age group. Clinical manifestations included chronic productive cough, hemoptysis, shortness of breath, chest pain, non-specific fever, night sweats, fatigue, and weight loss that persisted for months to years. In CPH, focal or diffuse infiltrates, nodules, consolidation, and cavities (usually apical) are typical, and therefore difficult to distinguish from pulmonary TB. Interstitial fibrosis and pleural thickening may be observed at the later stages [12,14]. In this study, we found most patients to have moderately or far advanced chest X-ray abnormalities. Patients with chronic disseminated histoplasmosis tend to have less active fungicidal activity against *H. capsulatum* in their macrophages [26,41,42]; perhaps this true of CPH as well. The role of diabetes mellitus as a risk factor for histoplasmosis has been noted but is not well understood; we found 7 of 22 (31.8%) patients positive for *Histoplasma* antibodies to be diabetic [43].

If it is related to gender, it appears that there is a strong tendency for men to experience more severe lung damage than women, especially in the group of patients with positive *Histoplasma* antibodies. (Table 5). The group of patients with mixed antibodies against *Histoplasma* and *Aspergillus* confirms this gender influence. Statistical calculations were not carried out due to the limited number of observations.

The older literature mostly supported CPA as a post-TB treatment infection, but more recent data shows both co-infection and misdiagnosis of de novo CPA as pulmonary TB. Pulmonary aspergillosis may occur in the presence of TB infection, as it relatively frequently does in non-tuberculous pulmonary infection. Lung disorders, especially those leading to cavities can facilitate the colonization of *Aspergillus*, including development of a fungus ball or aspergilloma, often a late feature of CPA. The number of TB cases in Indonesia is still very high and therefore the potential for TB pulmonary aspergillosis is also high; we estimated a prevalence of 387,700 cases based on an 8% incidence occurring at the end of anti-tuberculous therapy [11] and an annual 6.5% annual rate for 5 years in those left with pulmonary cavities after TB cure [44,45]. Chronic pulmonary aspergillosis in TB cases and ex-TB cases who repeatedly cough up blood has been reported sporadically in Indonesia [46]. As with CPH, we found 62.5% of those with positive *Aspergillus* antibodies to be diabetic, compared with 29.4% in those without.

We found two surprising associations with fungal antibody positivity. *Histoplasma capsulatum* is a dimorphic fungus that thrives on soil contaminated with bird and bat feces and is endemic to humid climates. People become infected with microconidia inhalation, especially in enclosed environments like caves, farms, silos, dovecotes, chicken hatcheries [47]. This study found that living in a humid environment was a risk factor for histoplasmosis in MDR-TB patients, where there was twice the risk of histoplasmosis when living in humid spaces compared to living in dry spaces. According to Menges-Robert et al., *H. capsulatum* grows best between 20 and 30 °C and on damp bark or soil [48]. Fungal spores remain viable for 6 to 8 weeks at 5 °C [44]. These findings are consistent with our novel association. We also found that people who have pets are 18 times more likely to have *Aspergillus* seropositivity than people who do not have pets. Allergic aspergillosis was more common in cystic fibrosis patients with frequent pet contact, but this association with CPA has not been a previously noted [49,50].

The limitations of this study include the relatively small sample size, the lack of fungal culture of the sputum, the intrinsic difficulty of interpreting radiological findings in patients with dual or triple lung infection and the less than perfect diagnostic performance of *Aspergillus* and *Histoplasma* antibody testing. A general limitation to the diagnosis of CPH is the lack of a standardized definition and few data on characteristic radiological findings.

However, our data indicate that a search for fungal co-infection is called for in MDR-TB cases, and this can be achieved simply by detecting serum antibodies, given that fungal culturing is insensitive and *H. capsulatum* is a class 3 pathogen. Much more work remains to define the full implications of our observations. We know that CPA carries a poor prognosis and needs treatment in most cases, progressing slowly and if left untreated, it may be fatal [48,51].

## 5. Conclusions

This study found that one third of MDR-TB patients have anti-*Aspergillus* or anti-*Histoplasma* IgG antibodies or both, consistent with concurrent or sequential CPA and CPH. If confirmed and replicated in other locations, our findings have profound significance for the management of MDR-TB.

## Figures and Tables

**Figure 1 jof-10-00529-f001:**
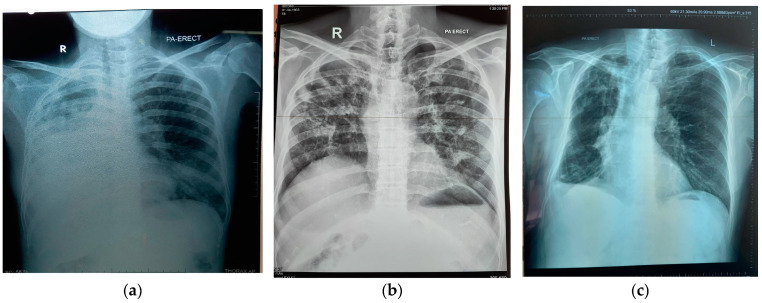
Chest X-ray examples in cases of chronic pulmonary histoplasmosis and aspergillosis: (**a**,**b**) far advanced radiological findings, (**c**) moderately advanced radiological findings.

**Figure 2 jof-10-00529-f002:**
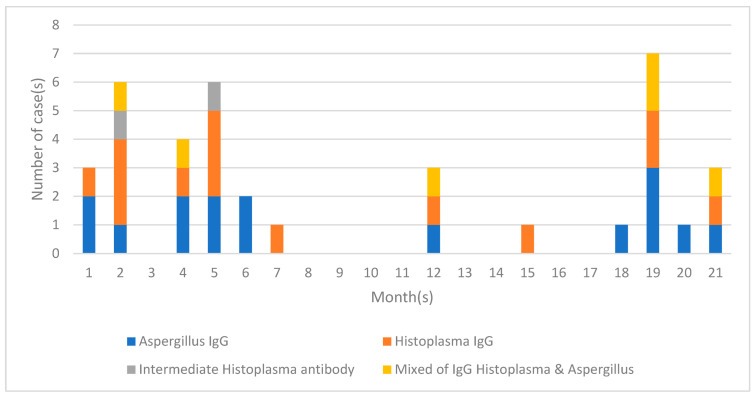
Sampling interval between the diagnosis of MDR-TB and the detection of *Aspergillus* and *Histoplasma* antibodies. Those patients who were positive for antibodies could be divided into three clusters, i.e., early (1–7 months), medium (8–14 months) and late-onset (15–21 months) of fungal infection, but sampling was only carried out on one occasion.

**Table 1 jof-10-00529-t001:** Distribution of patients with positive and negative anti-*Histoplasma* [16] and anti-*Aspergillus* antibody results.

Variable	Total (*n* = 50)	%
Positive *Histoplasma* IgG	14	28%
Negative *Histoplasma* antibodies	34	68%
Indeterminate result of *Histoplasma* antibodies	2	4%
Positive *Aspergillus* IgG/IgM	16	32%
Negative *Aspergillus* antibodies	34	68%
Mixed positive *Histoplasma* and *Aspergillus* antibodies	6	6%
Both were negative	24	48%

**Table 2 jof-10-00529-t002:** Serological status and patient demographics.

	Characteristic	Positive *Histoplasma* Antibody *n* = 14	Negative *Histoplasma* Antibody * *n* = 36	Positive *Aspergillus* Antibody *n* = 16	Negative *Aspergillus* Antibody *n* = 34
Gender	Male	10	26	10	26
	Female	4	10	6	8
Age group	16–30 years old	0	7	0	7
	31–45 years old	3	13	3	14
	46–60 years old	8	13	9	12
	>60 years old	3	3	4	2
Comorbidity	Diabetes mellitus	7	15	10	10
	HIV	0	2	0	0
	Hypertension	2	1	1	2
	Asthma	0	1	0	1
	Cancer	0	1	0	1
	No comorbidity	7	18	6	21
Smoking history	Smoker	0	0	0	0
	Ex-smoker	5	10	2	13
	Non-smoker	9	26	14	21
History of working	Yes	3	8	4	7
in a farm or plantation	No	11	28	12	27
History of having a poultry	Yes	4	11	7	8
	No	10	25	9	26
Humid living environment	Yes	5	19	6	18
	No	9	17	10	16
History of TB treatment	Yes	10	22	11	22
	No	4	14	5	12
MGIT culture **	Positive	4	13	10	24
	Negative	10	23	6	10
Lesion area on the chest X-ray	Minimally advanced	1	14	4	11
	Moderately advanced	5	11	4	12
	Far advanced	8	11	8	11

Part of Table 2 is modified from Khairunisa et al. [16]. * Two patients had intermediate results in the *Histoplasma* antibody detection. ** Mycobacteria growth indicator tube.

**Table 3 jof-10-00529-t003:** Multivariate analysis of all determinants of a positive *Histoplasma* IgG antibody assay.

Determinants	Prevalence Rate	*p* Value	95% C.I. for PR	
			Lower	Upper
Plantation/farm	0.906	0.904	0.182	4.510
Pets	0.979	0.977	0.229	4.181
Humid houses	2.797	0.226	0.530	14.762
Smoking history	0.403	0.304	0.071	2.284
Previous TB	1.070	0.924	0.266	4.308
Gender	1.366	0.709	0.266	7.012

**Table 4 jof-10-00529-t004:** Multivariate analysis of all determinants of a positive *Aspergillus* IgG/IgM antibody assay.

Determinants	Prevalence Rate	*p* Value	95% C.I. for PR	
			Lower	Upper
Pets at home	18.024	0.019	1.594	203.744
Gender	2.994	0.222	0.515	17.426
Previous TB	1.273	0.795	0.207	7.816
Plantation/farm	1.335	0.786	0.165	10.810
Humid houses	0.748	0.756	0.120	4.672
Acid fast result	0.063	0.064	0.003	1.178
Positive MGIT culture *	0.826	0.851	0.113	6.062
History of disease	4.180	0.135	0.641	27.237
Chest X-ray	4.545	0.028	1.180	17.514

* Mycobacteria growth indicator tube.

**Table 5 jof-10-00529-t005:** Severity of findings on chest X-rays according to *Histoplasma* and *Aspergillus* antibody positivity in male and female subjects.

Chest X-ray	*Histoplasma* Antibody +		*Aspergillus* Antibody +		Both Antibody +	
Findings	Male	Female	Male	Female	Male	Female
Minor changes	1	1	2	2	2	2
Moderately advanced	3		2	2	2	2
Far advanced	5	1	5	3	5	3

## Data Availability

The data is available in the corresponding author.

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
