# Peer review of "The Association of Chronic Pulmonary Aspergillosis and Chronic Pulmonary Histoplasmosis with MDR-TB Patients in Indonesia"

_jof, 2024, doi:10.3390/jof10080529_

Round 1

Reviewer 1 Report

The article “Chronic Pulmonary Aspergillosis and Histoplasmosis in MDR TB Patients in Indonesia”.

From my point of view, the manuscript is easy to follow and the subject is interesting. Noni N. Soeroso et al. investigated with whether MDR- TB patients (50) might have concurrent CPH or CPA, which could lead to a worse prognosis. Interesting study to be replicated in other regions and improve patient management, even complement with other diagnostic methods, not just detecting Histoplasma IgG antibodies and Aspergillus IgG/M antibodies.

Some points that should be clarified:   

·      - In the abstract, line 34: it indicates that 4 of 26 patients died…Could you clarify which are those 26 patients?.

·      - Lines 170, 173, 259 and 274: cite references in the text.

·      - Line 184: review the text. 

·    -The age of inclusion of study patients is indicated as being over 18. Table 2 indicates the range of patients from 16 years of age. Please explain it.

·      -It would be of great interest for the authors to evaluate the possibility of performing cultures, histopathology and/or molecular biology tests by PCR for the detection of the study fungi.

Author Response

Thank you to Reviewer -1 and Reviewer- 2 who have worked very hard to help the authors improve this manuscript

Comment 1:

  1. In the abstract, line 34: it indicates that 4 of 26 patients died. Could you clarify which are those 26 patients?

Answer-1. Thanks for the comments. We have stated it in Line 207-2015. We add an explanation about radiology images. Below is the changes: 

Line 207-2015: There were no deaths in the MDR-TB group who did not have antibodies against either H. capsulatum or Aspergillus fumigatus. In the group with positive H. capsulatum antibody one person died two months after diagnosis of MDR TB and a month after detection of Histoplasma antibodies. Two individuals in the group with positive Aspergillus antibody died two and five months after MDR-TB was diagnosed. Furthermore, one individual who had both antibodies also died three months after the diagnosis of MDR TB was established (Figure 2). All patients passed away during the treatment for MDR-TB; two of them had diabetes mellitus. All four patients had advanced radiological images by the established criteria, with the female patient who tested positive for both Aspergillus and Histoplasma antibodies having the most severe lung radiological images compared to the other three patients. That patient had extensive infiltrate in both lungs, and the right lung had collapsed (Figure 1a). Meanwhile, three other patients, although they had extensive infiltrates in both lungs, had no collapsed lungs. The proportion dying with positive fungal antibody dying was not significant compared with those without antibody (p = 0.11).

Comment 2: Lines 170, 173, 259 and 274: cite references in the text

Answer - 2: It has been done, please see in the manuscript

Comment 3. Line 184: review the text.  

Answer - 3,  We have erased the word Indonesia.

Comment 4. The age of inclusion of study patients is indicated as being over 18. Table 2 indicates the range of patients from 16 years of age. Please explain it.

Answer - 4: It is a typo - it's been corrected·  

Comment 5. It would be of great interest for the authors to evaluate the possibility of performing cultures, histopathology and/or molecular biology tests by PCR for the detection of the study fungi.

Answer - 5: We did not perform sputum culture because some patients found it difficult to collect phlegm, some patients passed away, and others were lost to follow-up after their antibodies were detected in the blood. This limitation hindered our ability to evaluate the possibility of conducting cultures, histopathology, and/or molecular biology tests using PCR to detect the studied fungi. We agree with the reviewer that molecular detection of Histoplasma in respiratory samples is likely to be rewarding and we are genuinely surprised that these data are missing from the literature.

Reviewer 2

Comment of Reviewer -2

Thank you for your nice comments

It is a very interesting study and made us aware to look for this association in other places of the world with high prevalence of MDR TB and CPA and CPH.

 Comment – 1 Very few (minor ) suggestions: Line 184: please correct this: Indonesia:  remove the line.  Please make sure that in the text: the names of microorganisms appear in italics: Aspergillus spp  Aspergillus spp: line 237. 

Answer - 1: Thank you. It was done.

Comment -2 Very important that the authors made the comment that the Histoplasma Ig G antibody assay has been evaluated only in the USA with a small number of CPH patients. Ideally, this test must be validated in other studies with more number of patients.  Also the comment of limitation of this study, that only blood was the sample cultured. 

Answer - 2: We have indeed made this comment on L303. This is a major deficiency of the literature as these tests are frequently done in clinical practice in the Americas, yet evaluations of Histoplasma IgG performance using in house of commercially available assays are almost completely missing.

Reviewer 2 Report

It is a very interesting study and made us aware to look for this association in other places of the world with high prevalence of MDR TB and CPA and CPH.

Very few (minor ) suggestions: Line 184: please correct this: Indonesia:  remove the line.  Please make sure that in the text: the names of microorganisms appear in italics: Aspergillus spp=  Aspergillus spp: line 237.

Very important that the authors made the comment that the Histoplasma Ig G antibody assay has been evaluated only in the USA with a small number of CPH patients. Ideally, this test must be validated in other studies with more number of patients.  Also the comment of limitation of this study, that only blood was the sample cultured.

It is a very interesting study and made us aware to look for this association in other places of the world with high prevalence of MDR TB and CPA and CPH.

Very few (minor ) suggestions: Line 184: please correct this: Indonesia:  remove the line.  Please make sure that in the text: the names of microorganisms appear in italics: Aspergillus spp=  Aspergillus spp: line 237.

Very important that the authors made the comment that the Histoplasma Ig G antibody assay has been evaluated only in the USA with a small number of CPH patients. Ideally, this test must be validated in other studies with more number of patients.  Also the comment of limitation of this study, that only blood was the sample cultured.

Author Response

Thank you for helping us improve our manuscript

Comment of Reviewer -2

Thank you for your nice comments

It is a very interesting study and made us aware to look for this association in other places of the world with high prevalence of MDR TB and CPA and CPH.

 Comment – 1 Very few (minor ) suggestions: Line 184: please correct this: Indonesia:  remove the line.  Please make sure that in the text: the names of microorganisms appear in italics: Aspergillus spp  Aspergillus spp: line 237. 

Answer - 1: Thank you. It was done.

Comment -2 Very important that the authors made the comment that the Histoplasma Ig G antibody assay has been evaluated only in the USA with a small number of CPH patients. Ideally, this test must be validated in other studies with more number of patients.  Also the comment of limitation of this study, that only blood was the sample cultured. 

Answer - 2: We have indeed made this comment on L303. This is a major deficiency of the literature as these tests are frequently done in clinical practice in the Americas, yet evaluations of Histoplasma IgG performance using in house of commercially available assays are almost completely missing.